Review

 

**Subject Area:**
biochemistry

human milk oligosaccharides, bacteria, microbiome, gut flora

**Author for correspondence:**
Steven D. Townsend
e-mail: steven.d.townsend@vanderbilt.edu

# Temporal development of the infant gut microbiome

Rebecca E. Moore and Steven D. Townsend

Department of Chemistry, Vanderbilt University, Nashville, TN 37235, USA

SDT, 0000-0001-5362-7235

The majority of organisms that inhabit the human body reside in the gut. Since babies are born with an immature immune system, they depend on a highly synchronized microbial colonization process to ensure the correct microbes are present for optimal immune function and development. In a balanced microbiome, symbiotic and commensal species outcompete pathogens for resources. They also provide a protective barrier against chemical signals and toxic metabolites. In this targeted review we will describe factors that influence the temporal development of the infant microbiome, including the mode of delivery and gestational age at birth, maternal and infant perinatal antibiotic infusions, and feeding method—breastfeeding versus formula feeding. We will close by discussing wider environmental pressures and early intimate contact, particularly between mother and child, as they play a pivotal role in early microbial acquisition and community succession in the infant.

## 1. Introduction

The relationship between a human being and their microbiome is a mutualistic symbiosis wherein the human host provides nutrition and protection for the microbial community [1]. In turn, the microbial population assists with essential functions such as aiding in immune system development and providing defence against enteric infection [2]. Dysbiosis, or microbial imbalance, is linked to a number of diseases in infants such as asthma, Crohn's disease, inflammatory bowel disease (IBD), necrotizing enterocolitis and type 1 diabetes (T1D) [3–7]. While the resident microbes of the host flora are reasonably well characterized, the mechanisms and timing of inoculation are largely understudied. For the purposes of this review, we divide the development of the microbiome into three stages (figure 1).

*In utero* (or the *prenatal stage*) is the least understood period of microbial development [8]. The thought that the womb is sterile and, accordingly, that a neonate's microbiome is first seeded at birth is the accepted dogma. However, studies consistently emerge that suggest microbial communities exist in the placenta, amniotic fluid and meconium [9]. Accordingly, intrauterine seeding is an intriguing possibility. For example, it is believed that the placenta harbours a wide range of microbes, many of which originate in the mouth [10]. Thus, if children are exposed to a placental flora *in utero*, one can readily understand why maternal prenatal oral health is so important.

The next stage of flora development is parturition (or *labour and delivery*). Due to pop culture, the most famous part of this process is amniorrhexis (i.e. rupture of membranes or water breaking). Upon rupture of the amniotic sac, sterility is lost. As the baby descends through the birth canal, they experience their first wave of microbial inoculation via the vaginal flora [11]. These microbes deoxygenate the gut and set the stage for correct growth and development. As we characterize below, the mode of delivery drastically affects microbial colonization. Babies delivered via Caesarean section (CS) experience altered, less beneficial microbial inoculation [12]. This change in delivery mode

royalsocietypublishing.org/journal/rsob    Open Biol. 9: 190128

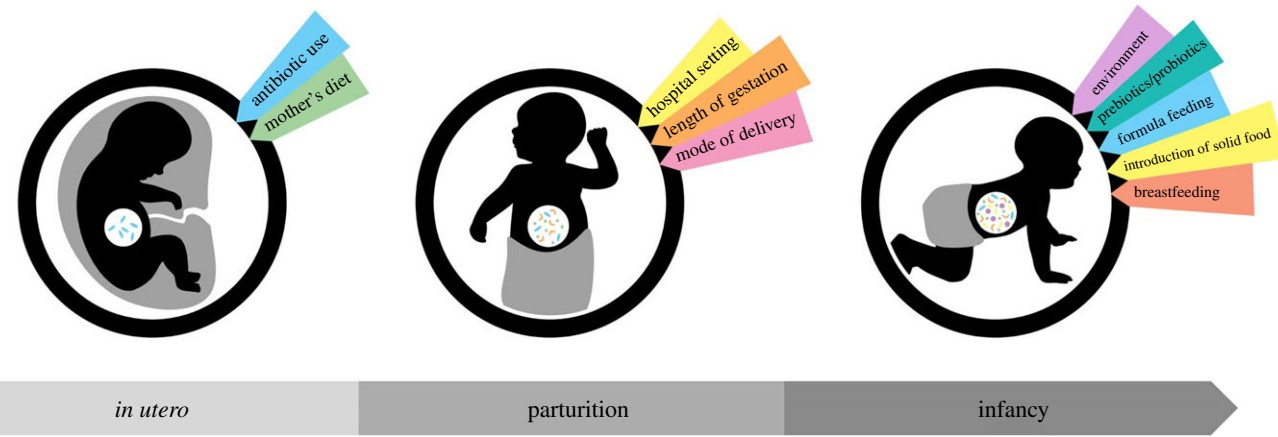

**Figure 1.** Stages and associated factors that modulate the microbiome early in life.

may ultimately explain why CS is associated with a number of long-term health challenges.

After parturition, the baby's microbial community experiences rapid changes. During infancy and the *postnatal stage*, skin-to-skin contact transfers valuable skin microbes to the baby [13]. A number of these organisms possess antimicrobial properties that defend against pathogens. While human skin is a valuable inoculator of the newborn, the most powerful and overwhelming source of microbes arrives via breast milk [14,15]. Due to its health benefits, the American Academy of Pediatrics and the World Health Organization (WHO) recommend exclusive breastfeeding through the first six months of life. In addition to helping a child achieve optimal growth and development, a primary consequence of consistent and exclusive breastfeeding is the proliferation of symbiotic and commensal gut microbes. Complex milk oligosaccharides, which are unique to primate milk, function as prebiotics (to provide a selective growth advantage for symbiotes over pathogens) [16–18], anti-adhesive antimicrobial agents (which selectively bind pathogens) [19], and antimicrobial and antibiofilm compounds (which are bacteriostatic against select pathogens) [20–25].

As solid foods are introduced into the diet, the microbiome begins the process of evolving from a simple environment that is *Bifidobacteria*-rich (microbes that metabolize human milk oligosaccharides) to a diverse flora rich in species such as *Bacteroides* that metabolize the starches present in a more complex diet. While the introduction of solid foods initiates microbial community changes, it is the gradual termination of breastfeeding that has the most profound effect. Thus, as weaning is initiated, a toddler's gut begins a maturation process leading to a diverse adult flora.

It is clear that stable microbial communities are established via dynamic changes during infancy. This review aims to characterize those changes at the molecular and microbial levels, to provide insight into the early development of the microbiome.

## 2. Prenatal development of the infant microbiome

The process of microbial colonization during early life is significant, as this time frame is critical to correct immunological and physiological development. Given the importance of commensal and symbiotic microbial communities to their

host's development, a number of mechanisms have evolved to facilitate their structured transmission. Microbial inoculation is well established in a number of host–microbial symbioses, where it ranges from vertical transmission from the mother to horizontal transmission from all other sources [26]. In contrast to these well-defined models of symbiosis, the mechanisms that drive the seeding of the more complex floras in and on the human body are poorly understood. Given the importance of the microbiota for proper human health and development, it is essential that we precisely characterize the source of microbes, the timing of colonization, and the endogenous and exogenous factors that govern these processes.

The current dogma is that the fetus develops in a sterile environment [9]. According to the sterile womb paradigm, microbes are acquired both vertically (from the mother) and horizontally (from the community) *during* and *after* birth. However, the degree of uterine sterility and the possibility of an *in utero* microbial community is highly contested. According to the *in utero* colonization hypothesis, microbial colonization of the human gut begins before birth. However, there are no rigorously conducted studies to support this hypothesis and challenge the sterile womb paradigm. Recent findings that suggests *in utero* colonization rely heavily on PCR and next-generation sequencing [27,28]. While compelling, each of these techniques lacks the detection limit required to study bacterial populations present in low quantities. Accordingly, based on the available data, a human being's first microbial inoculation occurs during labour and delivery. Many of these microbes have been characterized and quantified postpartum, *vide infra*. The earliest colonizers are adventitious species from the vaginal flora that typically gain penetrance to the digestive system through the baby's mouth. The nature of the microbes present is correlated to the health of the mother. Thus, conditions encountered during pregnancy or labour and delivery will influence the microbes that first colonize a child.

### 2.1. Maternal or neonatal antibiotic infusions

Over the last century, antibiotics have proven to be highly successful in treating bacterial infections. However, they are also well-known contributors to microbiome dysbiosis. Surprisingly, even though antibiotics are some of the most commonly prescribed medications to children, there are few studies detailing their long-term effects on the developing flora. This is neglectful when one considers that several

countries, including the United States, mandate that children receive antibiotic prophylaxis immediately after birth. Moreover, in Western countries, approximately 50% of women are exposed to an antibiotic during labour and delivery. For example, antibiotic use during pregnancy is standard during CS or assisted vaginal delivery. Before the procedure, the mother is given an intrapartum antibiotic prophylaxis (IAP) to help reduce the risk of CS related infections such as endometritis, urinary tract infections and surgical site infections [29]. While antibiotics can be delivered after the umbilical cord is clamped to reduce transmission to the newborn, the WHO recommends preoperative administration to lower the risk of post-CS maternal infections [30].

Mothers who plan on vaginal delivery do not require antibiotic administration, with the exception of those who test positive for group B streptococcus (GBS) [31]. GBS is a Gram-positive bacterium present in the GI tracts and genital tracts of 20–30% of pregnant women at the time of delivery [32]. Fifty per cent of all pregnant women will carry GBS at some point over the course of their pregnancy. While healthy pregnant women are typically asymptomatic, GBS infections in fetuses or infants are detrimental [33]. GBS is associated with preterm birth and infant mortality, and is a major cause of sepsis, pneumonia, meningitis and bacteraemia [33,34]. Additionally, there is a correlation between the timeline of antibiotic administration prior to vaginal delivery and the infant's microbiome make-up, with a decrease in *Bifidobacterium* and an increase in *Clostridium* [35].

The microbiome of CS-delivered infants is consistent with infants whose mothers were treated with IAP. The overall diversity of the neonatal gut is much lower. Typically, there are decreased levels of *Actinobacteria*, *Bacteroidetes*, *Bifidobacterium* and *Lactobacillus*, while there are increased levels of Proteobacteria, Firmicutes and Enterococcus spp. [35–37]. IAP administration has been linked to an elevated risk for the development of a number of diseases and conditions in the child including asthma, allergies and obesity. However, these studies are not all conclusive on whether they can be connected entirely to maternal antibiotic use [31,35].

While longitudinal studies are not available, what is known is that intrapartum antibiotic use is associated with decreased bacterial diversity in the neonate's first stool and a presumed lower abundance of lactobacilli and bifidobacteria in the gut. Similar associations have been observed after administration of antibiotics to the neonate directly after birth. Studies are unavailable that characterize the effect of prenatal antibiotics on the neonate's microbiome. Additionally, more data are needed that examine the potential effects of perinatal antibiotic use on an infant's short- and long-term health.

## 3. Development of the infant microbiome during parturition

Caesarean section delivery is a necessary surgical procedure when natural, vaginal delivery puts either the mother or baby at risk due to complications during the pregnancy or labour. The WHO recommends the rate of CS to be between 10% and 15%; however, the rate of caesarean deliveries is on the rise, especially in developed countries [38,39]. Despite

the potential risks involved for both mother and child, according to the Centers for Disease Control and Prevention (CDC), 32% of all deliveries in the United States are by CS [38,40]. There are several factors involved with making the decision to perform a CS with additional consideration made for elective CS. High-risk pregnancy conditions leading to CS include higher maternal age, obesity, pre-existing conditions such as diabetes, blood disorders and high blood pressure, and other factors such as multiple gestation, birth defects or pre-eclampsia [41,42]. Planned vaginal deliveries can lead to delivery by CS due to complications that arise during labour, necessitating emergency CS. The most common reasons for performing emergency CS are cephalo-pelvic disproportion (CPD), failed induction, macrosomia and non-reassuring fetal heart rate (NFHR) [43].

Aside from the numerous complications that can arise from CS, mode of delivery is one of the foremost contributors to disruption of the infant's microbiome; other contributors are maternal antibiotic use and formula feeding [35]. The development of the neonate's microbiota for those babies born vaginally is quite different from that of those babies born by CS. The infant is exposed to an expansive number of bacterial microbes through contact to vaginal, faecal and skin microbes following delivery (figure 2) [44,45]. Passage through the birth canal affords the neonate a microbiota similar to the mother's vagina, while CS babies' microbiota resembles the mother's skin and environmental microbes [45]. Broadly speaking, following birth, babies delivered by CS exhibit a decreased colonization of *Bacteroides*, *Lactobacillus* and *Bifidobacterium*, with an increased abundance of *Clostridium difficile* and common microbes associated with the skin such as *Staphylococcus*, *Streptococcus* and *Propionibacterium* [35,45,46]. There is a grey area with the differences in emergency versus planned CS, signifying that the onset of labour or membrane rupture can significantly alter the microbiota [46]. The microbial composition skews towards that of those delivered vaginally. While there is no clear answer to how long after birth the mode of delivery affects the microbiota of the child, the most significant differences are found up to 1 year after birth [47].

After the first week of life and up to 1 month of age, CS babies consistently showed significantly lower levels of *Bifidobacterium* and higher levels of *Klebsiella*, *Haemophilus* and *Veillonella* [48,49]. During the same time period, vaginally born infants displayed an increased abundance of *Bacteroides* [49]. After the first 30 days and up to 90 days after birth, species diversity between delivery modes is not as significant; however, several studies note that these differences are still detected. *Lactobacilli* and *Bifidobacteria* species are more abundant in vaginally delivered infants [50,51]. Within the Bacteroidetes phylum, the species diversity between CS and vaginally delivered babies is still present. CS delivered neonates typically display a lower abundance of *Bacteroides* and higher levels of *Enterobacteriaceae* and *Clostridium* [49,52].

By the age of 6 months, the colonization patterns are almost the same between the two modes of delivery; however, the *Bacteroides* and *Parabacteroides* species continue to be higher in vaginally delivered infants, while infants delivered by CS display a higher relative abundance of *Clostridium* spp. [31,47]. The variation in *Lactobacillus* colonization is no longer associated with delivery mode by the time the baby reaches 6 months of age [53]. Once the baby reaches a year

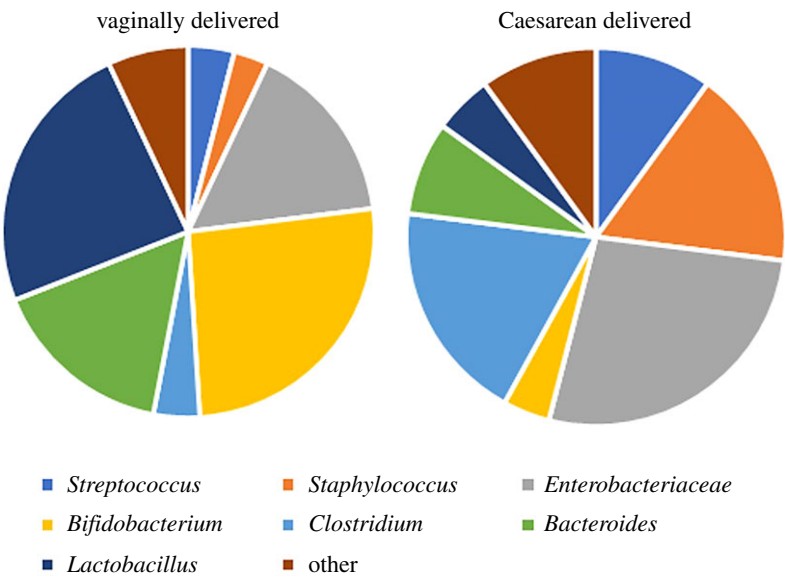

Open Biol. **9**: 190128

**Figure 2.** Comparison of bacteria present in the microbiomes of vaginally born and Caesarean-born infants. These ratios show the relative abundances of each species and combine the results of a number of studies.

of age, so many other factors are involved with the development of the microbiota of the baby that the differences are more difficult to attribute to the mode of delivery. The *Bacteroides* continued to appear in relatively lower abundances in CS delivered infants as well as a lower diversity in the species within the Firmicutes and Proteobacteria phyla [52].

The risks of developing immune-associated and allergic diseases, as well as hard to treat infections, are much higher following CS delivery. Conditions connected to delivery by CS include inflammatory bowel disease, T1D, coeliac disease, childhood asthma and obesity [54]. While it is clear that delivery by CS contributes to infant gut dysbiosis, there are strategies in place to help shape infants' gut microbiota to a normal composition and decrease the risk of infection and disease. In order to counteract the lack of exposure to the maternal vaginal community, a process termed 'vaginal seeding' can be carried out. About 1 h before the CS surgery, a sterile pad soaked in saline is inserted in the mother's vagina, provided she tests negative for GBS and vaginosis [55,56]. Within minutes of Caesarean birth, the infant is inoculated with the mother's vaginal fluid through a swab of gauze to the mouth, face and body [56]. Another common strategy to reduce the risk of immune-related diseases caused by dysbiosis of the microbiome of CS-delivered babies is to administer probiotics to high-risk pregnant women [57]. In addition, infant formula or probiotic drops are often supplemented with *Lactobacillus reuteri* to help lessen the effects of infant gut disorders [57].

## 4. Influence of feeding on the infant microbiome during the postnatal period

### 4.1. Human milk versus infant formula

The numerous benefits that breastfeeding provides both short-term and long-term for the child are well known. This is why the WHO recommends exclusively breastfeeding for the first 6 months of life, followed by supplemental breastfeeding up to 2 years as solid foods are introduced [58]. Breast milk provides protective measures against the risk of acquiring infectious diseases and developing atopic disorders through its immunological components including immunoglobulins, cytokines, growth factors and microbiologic factors. These components are especially important for the growth and development of the young infant's immune system [59].

The composition of the milk is dynamic, changing throughout lactation to satisfy the needs of the infant at different stages of its development, especially during the first few weeks (table 1). The colostrum is the first milk produced after birth. It is a thick, yellow fluid, and while it lacks high nutritional value, it is rich in immunologic and growth factors [60]. Colostrum production begins mid-pregnancy and, at about 5 days postpartum, slowly transitions to traditional breast milk over a period of 2 weeks. By 4 weeks postpartum, the milk is considered fully mature with limited changes in composition throughout the remaining lactation period.

Immunoglobulins (Ig) are important components that protect the neonatal gut against pathogenic bacteria. The immunoglobulins found in human milk include IgA, secretory IgA (SIgA), IgM, secretory IgM (SIgM) and IgG, with SIgA playing a central role in its defence against infectious disease [59]. While SIgA is present throughout all periods of breastfeeding, it is found in its highest concentrations in the colostrum [61]. SIgA works through binding to pathogens in the intestinal lumen, preventing their attachment to epithelial cells and mucosal regions [62,63].

Cytokines are secreted proteins found in human milk that function in developing the infants immune system through their anti-inflammatory and immunosuppressive properties [59,64]. The variety and concentrations of the individual cytokines vary from mother to mother and throughout the lactation period. However, interleukins-6, 8 and 10 (IL-6, IL-8 and IL-10), tumour necrosis factors-α and β (TNF-α and TNF-β), and transforming growth factors-α and β (TGF-α and TGF-β) are commonly found across lactating mothers [59,65]. IL-6 is

**Table 1.** Summary of significant components found in human milk.

| component | function |
| --- | --- |
| immunoglobulins | prevent pathogens from entering into systemic circulation and binding to epithelial surfaces |
| cytokines | anti-inflammatory and immunosuppressive agents |
| growth factors | modulate metabolic system development (digestive, nervous, etc.) |
| lipids | participate in nervous system and gastrointestinal development provide protection against enteric infection |
| proteins | nutrition, nutrient absorption, antimicrobial activity, gut and immune system development |
| human milk oligosaccharides (HMOs) | prebiotics, anti-adhesive, antimicrobial and antibiofilm agents |
| probiotics | consistent wave of commensal and symbiotic organisms |

royalsocietypublishing.org/journal/rsob    Open Biol. **9**: 190128

involved with the biosynthesis of IgA cells in the mammary glands; the highest levels are found in the colostrum [59,64]. Typically, IL-6 is found in higher concentrations in the mothers who deliver pre-term, indicating IL-6 is necessary to counteract the developmentally suppressed immune systems common among preterm babies [59].

The presence of a wide array of growth factors in human milk is especially important during the first weeks of life. These factors are responsible for the growth and development of a number of systems. The epidermal growth factor (EGF) is first present in the amniotic fluid and following birth, found both in the colostrum and mature milk. In the infant gut, EGFs promote cell proliferation and maturation of epithelial cells, as well as participate in intestinal mucosa repair [60,66]. The neuronal growth factors (NGFs) are involved in growth and development of the nervous system with a focus on both prenatal and postnatal brain maturation [60,67]. Erythropoietin (Epo), found in high concentrations in human milk, is a hormone involved in intestinal development and increased production of red blood cells, which in turn decreases the risk of anaemia [59].

In addition to the bioactive molecules present in human milk, the largest proportion of solid components includes the complex proteins, lipids and carbohydrates. The composition can be dynamic across mothers as the milk matures from the initial colostrum, but the average milk supply contains 3–5% fats, 0.8–0.9% proteins and 6.9–7.2% carbohydrates, with an additional 0.2% mineral component [68]. Milk fats make up 40–55% of the total energy in breast milk, with lactose providing an additional 40% [69,70]. Over 200 different fatty acids have been identified in human milk, with triglycerides accounting for over 98% of the fat content [69,71]. Complex lipids play a central role in brain and gastrointestinal development, as well as protection against pathogenic bacteria, specifically GBS [71].

With over 400 unique proteins found in human milk, the most common are casein, α-lactalbumin, lactoferrin, immunoglobulin IgA, lysozyme and serum albumin [71]. Milk proteins play a role in development of the neonatal gut and immune system, assist in nutrient absorption, and protect against pathogens through their antimicrobial activity [67,72]. Lactoferrin is found in high abundance in human milk and is known for its antibacterial activity against pathogens that gain virulence through an iron-mediated mechanism [73]. Lactoferrin has the ability to bind to two ferric ions, which is how it is thought to inhibit bacterial pathogens; however,

lactoferrin has also shown antimicrobial activity against non-iron requiring viruses and bacterial species [73,74].

While lactose is the most abundant carbohydrate found in human milk, human milk oligosaccharides (HMOs) are of the most interest when discussing the infant microbiome. HMOs are the third largest component in human milk, and, while infants are unable to digest them, they play an important role in shaping the microbiota of the developing gut and building up the young immune system. Over 200 unique HMOs have been identified, ranging from 3 to 22 sugars per molecule [71]. All HMOs are composed of five glycans: L-fucose, D-glucose, D-galactose, $N$-acetylglucosamine and $N$-acetylneuraminic acid [71]. Since they are relatively unaffected by digestion, HMOs are able to pass through the infant's stomach and small intestine intact, and they accumulate in the colon [75]. One of the primary functions of HMOs is to act as a prebiotic, allowing the growth of beneficial *Bifidobacterium* spp., while preventing the colonization of harmful pathogens.

In addition to the prebiotic nature of HMOs, breast milk is known to have probiotic properties that help to shape the infant gut microbiota. Once considered sterile, breast milk is actually the source of the $10^4$–$10^6$ bacterial cells per day that the infant consumes, with an average feeding of 800 ml per day [76]. While the source of the bacteria present in human milk is not completely clear, it is thought it is a combination of bacteria from the infant's oral cavity and from the mother's nipple and surrounding skin [75,77]. In exclusively breastfed infants, the most abundant bacterial genera are *Bifidobacterium*, *Lactobacillus*, *Staphylococcus* and *Streptococcus*. *Bifidobacterium* species dominate 70% of the strains [75,78]. The *Bifidobacterium* species most frequently detected that act to establish healthy gut flora are *B. breve*, *B. longum*, *B. dentium*, *B. infantis* and *B. pseudocatenulatum* [79,80].

For the first 6 months of the infant's life, when breast milk is typically the sole source of nutrition, the gut bacteria present varies significantly depending on the mother. While the introduction of solid foods does begin to create uniformity across the microbiome, there are still higher abundances of *Bifidobacterium* and *Lactobacillus* species present in breastfed babies [81]. It is not until breastfeeding ceases that the child's microbiome begins to resemble an adult-like state.

There are a number of factors that can lead to the introduction of formula supplementation into a baby's diet; however, only a limited number of neonates require formula for medical reasons. Social concerns and lack of prenatal breastfeeding education contribute to formula feeding,

along with insufficient milk production due to limited mother-to-infant contact, worry over the baby not receiving enough milk, trouble calming a fussy baby, and a lack of sleep. In fact, 85% of mothers plan to exclusively breastfeed for the first 6 months. However, less than 50% breastfeed exclusively at 3 months and about 25% at 6 months [82,83]. According to the CDC, as of 2015, 17.2% of infants receive formula supplementation within the first 48 h [83]. During the first days postpartum, only a limited supply of milk is produced before the onset of lactogenesis at 2–4 days [84,85]. This is a critical period in which mother and infant separation can delay the establishment of breastfeeding or even hinder the process from starting. And while breast milk is considered the best nutrition for the baby during the first 6 months of life, the advancement of the nutritional contents in formula over the past several decades has made formula a healthy alternative.

The goal of formula design is to promote growth and development of the infant through a product that mimics the nutritional composition of human milk. This is a difficult task due to the complexity of breast milk and the changes that occur in the nutritional profile, especially the macronutrients, throughout the course of lactation. While formula companies do their best, it is not feasible to include some of the components, such as the bioactive materials found in human milk, in their formula. Infant formula is government regulated to ensure the proper composition of proteins, fats, sugars, vitamins and minerals [70]. Cow's milk and soy milk are the two most common bases for infant formula. However, there are several additional specialized formulae available on the market to meet the needs of babies with certain sensitivities. Due to the high abundance of fats and proteins in cow's milk, it first must be diluted to a similar composition to human milk [70]. For babies with either colic or milk allergies, soy-based formulae are a common substitute.

Infant formulae are commonly fortified with prebiotics and/or probiotics to incorporate some of the beneficial components found in human milk [16,86]. Prebiotics are nondigestible oligosaccharides that stimulate the growth of beneficial bacteria in the digestive system. The most common oligosaccharides supplemented into formula are short-chain galacto-OS (GOS), long-chain fructo-OS (FOS) and polydextrose [87,88]. These oligosaccharides have been shown to stimulate the growth of beneficial *Bifidobacterium* and lower the abundance of *E. coli* and *Enterococcus* [87–89]. Probiotics are non-pathogenic, live microbial organisms that promote the growth of beneficial flora such as *Bifidobacterium* and *Lactobacillus* spp. [90]. The addition of these probiotics has been known to reduce the susceptibility of antibiotic-associated diarrhoea and the symptoms of colic [91,92].

The microbial composition of the infant gut of formula-fed babies is vastly different from that of breastfed babies. It has been shown that, even for mixed-fed babies (those who supplement formula between breast feedings), the gut microbiota more closely resembles the patterns of exclusively formula fed babies [82]. The infant microbiome is shifted towards that of an adult at a quicker rate with higher overall bacterial diversity [82]. The gut is dominated by *Staphylococcus*, *Streptococcus*, *Enterococcus* and *Clostridium* species, as well as specific species of *Bifidobacterium* [93,94]. In addition, in exclusively formula-fed babies, a greater prevalence of *E. coli*, *C. difficile*, *B. fragilis* and *Lactobacilli* species has been observed to colonize the gut [75,95].

## 4.2. Introduction to solid foods

The process of weaning, when the introduction of solid foods begins, typically starts at around 4–6 months and continues until the baby is approximately 2 years old. The WHO recommends complementary feeding should be gradual, in which the baby is still feeding off of either infant formula or breast milk as 'family foods' are supplemented into the diet [96]. Weaning becomes necessary when breast milk or infant formula no longer provides the necessary nutrients for the baby. Typically, at around 6 months of age babies begin to show signs they are ready by losing interest in nursing or becoming more interested in eating solid foods. According to the CDC, solid foods should be introduced one at a time to ensure that none of the common allergies to milk, eggs, fish, shellfish, tree nuts, peanuts, wheat or soya beans are present [97].

The weaning process must progress in a timely manner to ensure that the digestive system matures properly. Pancreatic function, small intestine absorption and fermentation capacity are underdeveloped during the early weaning stages [98]. While there are enzymes present in the saliva that help break down food, it is not until 6 months of age that the pancreas secretes enough enzymes including α-amylase to digest starches and proteins [99]. Until the pancreas gains full function, there are a vast number of non-digestible carbohydrates that are absorbed by the colon and allow the growth of beneficial bacteria that are unable to proliferate in a breast milk or formula diet [98].

Iron and vitamin D are commonly supplemented into a weaning infant's diet to reduce the risk of deficiencies. Babies are born with a supply of iron from their mothers; this iron is depleted by approximately 6 months of age [100]. At this point, babies must consume either an iron-fortified infant formula or eat a diet of iron-containing foods. Iron is essential for humans to synthesize haemoglobin, which is required to transport oxygen from the lungs to all the other cells in the body. When there is an iron deficiency, there are significant differences in the microbial environments of the infant gut. While most bacteria require iron for survival, growth and proliferation, in iron-low conditions, there is an increase in *Bifidobacterium* and *Lactobacillus* spp., which have little to no need for iron [101]. Vitamin D not only is crucial for building strong bones, but plays a role in the maturation of the gut microbiome. Vitamin D is enriched in breast milk and infant formula; however, during weaning, babies should be consuming vitamin-D-fortified milk, yogurt and cereals.

While it was previously thought that the introduction of solid foods alters the gut microbial composition, it is the cessation of breastfeeding that is now attributed to shifting the microbiota to an adult-like state. In general, the introduction of solid foods alters the gut microbiota to be dominated by bacteria within the Bacteroidetes and Firmicutes phyla. At the genus level, there is an increase in *Atopobium*, *Clostridium*, *Akkermansia*, *Bacteroides*, *Lachnospiraceae* and *Ruminococcus* spp.; concurrently there is a decrease in *Escherichia* and *Staphylococcus* spp. [75,102]. During the initial weaning period, when breastfeeding is still the sole source of nutrition, *Bifidobacterium* and *Lactobacillus* spp. continue to dominate at consistent amounts [75,103]. Even after several months of complementary feeding, these infants displayed lower abundances of *Clostridium leptum*, *Clostridium coccoides* and

royalsocietypublishing.org/journal/rsob    Open Biol. 9: 190128

*Roseburia* spp. There is an increase in the abundance of *Bifido-bacterium*, *Lactobacillus*, *Collinsella*, *Megasphaera* and *Veillonella* compared with those who ceased breastfeeding [75,81]. Formula-fed babies during the same period exhibit higher abundances of *Bacteroides*, *Clostridium difficile*, *Clostridium perfringens* and *Clostridium coccoides*, with overall less mature microbiota [98].

# 5. Environmental exposure

In addition to the obvious determinants of infant microbiome diversity, including mode of delivery, breastfeeding versus formula feeding, antibiotic use and introduction of solid foods, environmental exposures can also play a key role in the variability of that microbiota. Hospital setting, cohabitation with family members, geographical location, air quality, pet and animal exposure, and daycare are all included in the environment factors that contribute to neonatal microbiome development. A focus on neonatal intensive care unit (NICU) hospital environment and exposure to pets is examined in this review since there exists exhaustive research in this area.

One of the earliest routes of microbial transmission is in the hospital with the majority of births occurring in the setting or quickly being transferred there. Specifically, the NICU is associated with increased exposure to a diverse microbial community. While not an exhaustive or conclusive list, some of the common causes of preterm labour are maternal ethnicity, body mass index and age, infection or inflammation, smoking, and stress [104]. Due to the nature of the preterm birth, infants are often born spontaneously vaginally or through emergency CS. The mortality and infection rates of preterm infants are much higher than their full-term counterparts because their compromised immune systems make them more prone to NICU-acquired infections [105]. Even with intensive sanitization and care to keep the NICU a sterile environment, a plethora of both pathogenic and commensal bacterial species are found both on surfaces and elsewhere within the facility. The most common bacteria that are found to colonize on neonate-associated surfaces, which includes ventilators, CPAP machines, stethoscopes, feeding tubes, catheters and pacifiers, are *Streptococcus*, *Staphylococcus*, *Neisseria*, *Pseudomonas* and *Enterobacteriaceae* species [106,107]. The most prevalent environmental bacteria found within the NICU include *Geobacillus*, *Halomonas*, *Shewanella*, *Acinetobacter* and *Gemella* species [106,108].

The correlation between the infant gut microbiota and the bacterial species present in the NICU environment is evident through stool evaluation. In general, *Clostridia* species (specifically *C. perfringens*, *C. butyricum*, *C. difficile* and *C. paraputrificum*) are found in high abundances in the infant microbiota [109–111]. *Staphylococcus* species are found to be highly abundant in very-low-birthweight (VLBW) NICU infants, especially on the skin, but are also present in the gastrointestinal tract [110,112]. The stools of the VLBW neonates are typically colonized by *Klebsiella*, *Enterobacter* and *Enterococcus* species, while *Streptococcus* species dominate the saliva [108,112–114]. This is compared with normal-birthweight, healthy infants in which *Escherichia*, *Bifidobacterium* and *Bacteroides* species are found in higher abundances [108,110,112].

Early exposure to pets and other animals has been known to play a pivotal role in shaping the gut microbiome and is associated with a higher immunity and lower prevalence of allergy and asthma development. The microbial composition of homes with pets is examined through vacuumed house dust [35]. In general, homes with pets show lower abundances of *Bifidobacteriaceae* and higher abundances of *Peptostreptococcaceae* [115,116]. When compared with pet-free homes, infants are more likely to be colonized by *Bifidobacterium* species *pseudolongum*, *thermophilum* and *longum*. In one study *B. longum* was associated with a protective effect against bronchitis [35,115,117]. In another study with mice, there was found to be an increase of *Lactobacillus johnsonii*, providing a probiotic type of effect on the gut microbiota [118].

# 6. The TEDDY study

Gaining insight on the development of the microbiome from *in utero* through infancy into childhood is an important tool in helping us understand what role bacteria plays in human disease. One study, titled The Environmental Determinants of Diabetes in the Young (TEDDY), aimed to find a correlation between the early life factors that shape the infant gut microbiome and the risk of acquiring T1D. Two separate TEDDY studies have analysed the stools of children from around 3 months of age to the clinical endpoint (about 46 months of age) [119,120].

The key takeaway from the first study, which sequenced 12 500 stool samples from 903 children, concluded there are three stages of gut microbiome development: the developmental phase (3–14 months of age), the transitional phase (15–30 months of age) and the stable phase (31–46 months of age) [120]. This study was able to track the bacterial make-up throughout the entire 43-month time frame; attribute changes in the gut flora to either environmental, maternal or postnatal effects; and analyse whether or not this information could predict the onset of T1D [120]. Consistently in this study an increased abundance of *Bacteroides* and lower abundances of short-chain fatty acid producing bacteria were found [120].

The second TEDDY study analysed 10 913 stool samples from 783 children and followed a similar protocol to the first study [119]. Data were collected until either islet immunity was reached or the child persisted to test positive for T1D. The key takeaway from this study was that specific strains of bacteria may not be the cause of T1D, as was thought in the first study. In healthy control children, researchers found that the children's bodies contained more genes that are associated with fermentation and the biosynthesis of short-chain fatty acids [119]. This suggests that the short-chain fatty acids, instead of bacteria, are more vital in protecting the body against the onset of T1D [119].

While this comprehensive review on the development of the gut microbiome only focuses on the first year of life, it is important to recognize that the microbiome continues to mature after this critical stage. These two TEDDY studies together, while not conclusive, have laid the foundation for a necessary understanding of how the make-up of the microbiome can predict human disease.

# 7. Conclusion and future outlook

The early microbiome appears to follow a progression from organisms that facilitate lactate utilization during strict

royalsocietypublishing.org/journal/rsob    Open Biol. 9: 190128

lactation to anaerobic organisms involved in the metabolism of solid foods containing complex starches, once introduced. At around 12 months old, the infant microbiome achieves a more complex structure, and becomes similar to that of adults by the age of 3 years. In addition to facilitating nutrient usage, the ecological succession of the infant microbiome educates the immature immune and metabolic systems. Disruption of what has evolved to be a 'normal' assembly process may have considerable downstream consequences for the development of autoimmune and metabolic pathologies.

Going forward, governing the infant microbiome will almost certainly focus on three areas. First is reducing the use of Caesarean delivery, as this practice is associated with alterations to the infant microbiome. The second major focus will be on decreasing the misuse and overuse of antibiotics during the perinatal period—particularly until we better understand the downstream effects of such treatments on the mother and child. The third area will probably focus on an increase in breastfeeding and using the components of human milk for novel food products and therapeutics [121].

Data accessibility. This article has no additional data.
Competing interests. We declare we have no competing interests.
Funding. This material is based upon work supported by the National Science Foundation under grant no. NSF-Career-1847804 to S.D.T.
Acknowledgements. Harrison Thomas is acknowledged for his assistance with artwork.

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
