## [Reviewer comments · Open Biology]

Review History

RSOB-19-0128.R0 (Original submission)

Review form: Reviewer 1

Recommendation

Major revision is needed (please make suggestions in comments)

Do you have any ethical concerns with this paper?

No

Comments to the Author

The manuscript by Moore and Townsend describes the impact of various factors on the gut flora development at early stages (prenatal, infant, postnatal) of infants. While the authors chose an interesting topic within the author's research area, there are several aspects that can be improved for better influence of this topic. My recommendation is major revision, please see my listed comments below.

1. Content. The author discussed fairly well on three stages of infants, however, they missed discussion on several important findings in some publications (Development of the Human Infant Intestinal Microbiota, Temporal development of the gut microbiome in early childhood from the TEDDY study, Temporal Development of the Infant Gut Microbiota in Immunoglobulin

E-Sensitized and Nonsensitized Children Determined by the GA-Map Infant Array, Development of the Pediatric Gut Microbiome: Impact on Health and Disease, et al.)

2. Content. The development of infant microbiome is also affected by physical environment (e.g. infant's room), persons (e.g. father/relatives) with daily contact, et al. The authors should convey a comprehensive impression regarding the development of gut microbiome.

3. Presentation. The authors spent much effort on the writing, however, neglected the importance of illustration. This manuscript only has 3 simple displaying items, which is not enough for a good presentation. Readers may get lost easily without summarizing tables/figures when reading the educational writing.

4. Minor issues. 4.1. Bold figure/table callout for review purpose. 4.2. Page 6 line 5, provide full name of CS for first appearance, check page 7 line 11. 4.3. Check reference style.

Decision letter (RSOB-19-0128.R0)

02-Aug-2019

Dear Dr Townsend,

We are pleased to inform you that your manuscript RSOB-19-0128 entitled "Temporal Development of the Infant Gut Microbiome" has been accepted by the Editor for publication in Open Biology. The reviewer has recommended publication, but also suggest some minor revisions to your manuscript. Therefore, we invite you to respond to the reviewer's comments and revise your manuscript.

Please submit the revised version of your manuscript within 7 days. If you do not think you will be able to meet this date please let us know immediately and we can extend this deadline for you.

1) A text file of the manuscript (doc, txt, rtf or tex), including the references, tables (including captions) and figure captions. Please remove any tracked changes from the text before submission. PDF files are not an accepted format for the "Main Document".

2) A separate electronic file of each figure (tiff, EPS or print-quality PDF preferred). The format

should be produced directly from original creation package, or original software format. Please note that PowerPoint files are not accepted.

3) Electronic supplementary material: this should be contained in a separate file from the main text and meet our ESM criteria (see <http://royalsocietypublishing.org/instructions-authors#question5>). All supplementary materials accompanying an accepted article will be treated as in their final form. They will be published alongside the paper on the journal website and posted on the online figshare repository. Files on figshare will be made available approximately one week before the accompanying article so that the supplementary material can be attributed a unique DOI.

Online supplementary material will also carry the title and description provided during submission, so please ensure these are accurate and informative. Note that the Royal Society will not edit or typeset supplementary material and it will be hosted as provided. Please ensure that the supplementary material includes the paper details (authors, title, journal name, article DOI). Your article DOI will be 10.1098/rsob.2016[*last 4 digits of e.g. 10.1098/rsob.20160049*].

4) A media summary: a short non-technical summary (up to 100 words) of the key findings/importance of your manuscript. Please try to write in simple English, avoid jargon, explain the importance of the topic, outline the main implications and describe why this topic is newsworthy.

Images

Data-Sharing

It is a condition of publication that data supporting your paper are made available. Data should be made available either in the electronic supplementary material or through an appropriate repository. Details of how to access data should be included in your paper. Please see <https://royalsocietypublishing.org/rsob/for-authors> for more details.

Data accessibility section

Sincerely,

The Open Biology Team
<mailto:openbiology@royalsociety.org>

Reviewer's Comments to Author:

Referee:

Comments to the Author(s)

The manuscript by Moore and Townsend describes the impact of various factors on the gut flora development at early stages (prenatal, infant, postnatal) of infants. While the authors chose an interesting topic within the author's research area, there are several aspects that can be improved for better influence of this topic. My recommendation is major revision, please see my listed comments below.

1. Content. The author discussed fairly well on three stages of infants, however, they missed discussion on several important findings in some publications (Development of the Human Infant Intestinal Microbiota, Temporal development of the gut microbiome in early childhood from the TEDDY study, Temporal Development of the Infant Gut Microbiota in Immunoglobulin E-Sensitized and Nonsensitized Children Determined by the GA-Map Infant Array, Development of the Pediatric Gut Microbiome: Impact on Health and Disease, et al.)

2. Content. The development of infant microbiome is also affected by physical environment (e.g. infant's room), persons (e.g. father/relatives) with daily contact, et al. The authors should convey a comprehensive impression regarding the development of gut microbiome.

3. Presentation. The authors spent much effort on the writing, however, neglected the importance of illustration. This manuscript only has 3 simple displaying items, which is not enough for a good presentation. Readers may get lost easily without summarizing tables/figures when reading the educational writing.

4. Minor issues. 4.1. Bold figure/table callout for review purpose. 4.2. Page 6 line 5, provide full name of CS for first appearance, check page 7 line 11. 4.3. Check reference style.

Author's Response to Decision Letter for (RSOB-19-0128.R0)

See Appendix A.

Decision letter (RSOB-19-0128.R1)

15-Aug-2019

Dear Dr Townsend

We are pleased to inform you that your manuscript entitled "Temporal Development of the Infant Gut Microbiome" has been accepted by the Editor for publication in Open Biology.

You can expect to receive a proof of your article from our Production office in due course, please

check your spam filter if you do not receive it within the next 10 working days. Please let us know if you are likely to be away from e-mail contact during this time.

Sincerely,

The Open Biology Team
mailto: openbiology@royalsociety.org

Appendix A

1. Content. The author discussed fairly well on three stages of infants, however, they missed discussion on several important findings in some publications (Development of the Human Infant Intestinal Microbiota, Temporal development of the gut microbiome in early childhood from the TEDDY study, Temporal Development of the Infant Gut Microbiota in Immunoglobulin E-Sensitized and Nonsensitized Children Determined by the GA-Map Infant Array, Development of the Pediatric Gut Microbiome: Impact on Health and Disease, et al.)

Response: Completed

2. Content. The development of infant microbiome is also affected by physical environment (e.g. infant's room), persons (e.g. father/relatives) with daily contact, et al. The authors should convey a comprehensive impression regarding the development of gut microbiome.

Response: Completed

3. Presentation. The authors spent much effort on the writing, however, neglected the importance of illustration. This manuscript only has 3 simple displaying items, which is not enough for a good presentation. Readers may get lost easily without summarizing tables/figures when reading the educational writing.

Response: Completed

4. Minor issues. 4.1. Bold figure/table callout for review purpose. 4.2. Page 6 line 5, provide full name of CS for first appearance, check page 7 line 11. 4.3. Check reference style.

Response: Completed